# Unsupervised Transfer Learning Method via Cycle-Flow Adversarial Networks for Transient Fault Detection under Various Operation Conditions

**DOI:** 10.3390/s24154839

**Published:** 2024-07-25

**Authors:** Xiaoyue Yang, Long Chen, Qidong Feng, Yucheng Yang, Sen Xie

**Affiliations:** 1School of Rail Transportation, Wuyi University, Jiangmen 529020, China; xyyeoh@outlook.com (X.Y.);; 2CRRC Guangdong Railway Vehicles Co., Ltd., Jiangmen 529100, China; fengqidong@crrcgc.cc; 3Institute of Intelligence Science and Engineering, Shenzhen Polytechnic University, Shenzhen 518060, China

**Keywords:** fault detection, Cycle-Flow adversarial network (CFAN), transfer learning (TL), traction control system (TCS), various operation conditions

## Abstract

The efficient fault detection (FD) of traction control systems (TCSs) is crucial for ensuring the safe operation of high-speed trains. Transient faults (TFs) can arise due to prolonged operation and harsh environmental conditions, often being masked by background noise, particularly during dynamic operating conditions. Moreover, acquiring a sufficient number of samples across the entire scenario presents a challenging task, resulting in imbalanced data for FD. To address these limitations, an unsupervised transfer learning (TL) method via federated Cycle-Flow adversarial networks (CFANs) is proposed to effectively detect TFs under various operating conditions. Firstly, a CFAN is specifically designed for extracting latent features and reconstructing data in the source domain. Subsequently, a transfer learning framework employing federated CFANs collectively adjusts the modified knowledge resulting from domain alterations. Finally, the designed federated CFANs execute transient FD by constructing residuals in the target domain. The efficacy of the proposed methodology is demonstrated through comparative experiments.

## 1. Introduction

High-speed trains have emerged as one of the most crucial components within intelligent transportation systems. Traction control systems (TCSs), serving as the core power systems for high-speed trains, are intricately linked to trains’ reliability and safety. However, they also represent a major source of faults in both long-term operation and harsh operating environments. Consequently, fault detection and diagnosis (FDD) has become an active area of research over the past few decades [1,2,3].

Currently, FDD methods for high-speed trains can be broadly categorized into three groups: model-based approaches, signal-based approaches, and data-driven approaches. Despite their accessibility and high efficiency in producing FDD results, establishing model-based methods is challenging due to practical uncertainties and complex designs. Signal-based methods exhibit limited effectiveness in detecting minor symptoms, particularly in dynamic scenarios [4].

In the meantime, due to the widespread deployment of sensors in complex systems, data-driven methods have been extensively advocated for accomplishing fault detection and diagnosis (FDD) tasks by effectively processing a massive volume of data [1,5,6,7]. In [8], the authors proposed a discriminative stacked autoencoder (D-SAE) network based on feature integration boosting for bearing fault diagnosis. This method mitigated the performance degradation and enhanced the generalization ability in various scenarios. Ref. [9] proposed an innovative fault detection (FD) method for bogie. In this study, a Monte Carlo-based perturbation technique is employed to amplify the distinction between unexpected faults and known ones. Consequently, the FD outcome for unexpected faults can be obtained using dropout-based Bayesian deep learning. The authors in [10] proposed a fault diagnosis method for braking friction based on a one-dimensional convolutional neural network (1DCNN) and the GraphSAGE network. This approach effectively addresses the challenge of imbalanced fault samples by considering the correlation between different fault features. In addition, ref. [11] presented an incipient FDD method for running gear systems that leveraged Hellinger distance and slow feature analysis.

The aforementioned FDD methods primarily address permanent faults (PFs) in mechanical components or systems. However, transient faults (TFs), as a type of incipient fault, have the potential to develop into PFs and are responsible for most failures observed in electronic devices such as power electronics, sensors, and traction control units (TCUs) within TCSs.

In the context of complex industrial systems, multiple fault detection and diagnosis (FDD) methods have been developed specifically for transfer functions (TFs) [12,13,14,15,16,17,18]. Ref. [12] assesses and demonstrates the ability of a bulk built-in current sensor’s (BBICS) architecture to detect multiple and simultaneous TFs for integrated circuits. Ref. [14] studies fault tolerance in switching reconfigurable nano-crossbar arrays, considering both TFs and PFs. In [15], an innovative ontology-based fault propagation analysis approach (ontologyFPA) is proposed to analyze transient fault propagation effects in networked control systems (NCSs). Ref. [17] presents a TF detection and classification approach in power transmission lines based on graph convolutional neural networks. In [18], an optimal fractional-order method is proposed for TF diagnosis, which suppresses background noise and amplifies the faulty part of the signal. Afterward, kurtosis and the fault duration time are applied to locate the fault component.

However, the methods mentioned above perform in static or one fixed operation condition, which are not involved in dynamic cases [4]. Different operation conditions may lead to significant distribution differences, which means that an intelligent FDD model trained on data under a certain operation condition is usually not applicable to other operation conditions [19,20]. Traditional deep learning approaches necessitate a plethora of samples from diverse operational conditions for effective model training. Conversely, a TCS typically operates under steady-state conditions, resulting in imbalanced distributions across various operation conditions [21].

The primary challenges in the field of TF detection encompass the following:TFs exhibit sporadic and stochastic behavior, leading to impermanent damage that disappears unpredictably.The distribution of samples across different operational conditions is imbalanced, particularly for faulty samples which are significantly underrepresented.The features of TFs are inherently weak and can easily be overshadowed by background noise, especially in dynamic scenarios.

These characteristics make TFs challenging to detect.

In this context, transfer learning (TL) has been extensively discussed for extracting latent feature information and achieving precise fault detection under dynamic operation conditions. TL aims to enhance the performance of target domains by leveraging the knowledge embedded in diverse but related source domains, thereby reducing the reliance on a substantial amount of target domain data for constructing target learners [21,22].

Several FDD methods with TL have been developed for electrical systems. Ref. [23] proposes an FD method for traction converter faults in traction drive systems. This method consists of a federal neural network based on a variational autoencoder (VAE), which can perform the FD task with performance degradation. The authors of ref. [24] developed a hierarchical method for transformer rectifier unit (TRU) fault diagnosis and a transfer learning-based fault diagnosis method without training new models for different TRUs. In [25], a novel transferrable open-circuit fault diagnosis method is proposed for insulated gate bipolar transistors in three-phase inverters, which can be applied to different systems with the same topology but different parameters. The authors of ref. [26] developed an adversarial-based deep TL model that can detect and classify short-circuit faults in DC microgrids without using historical fault data. Ref. [27] proposes a transfer learning-based fault location method for voltage source convertor-based high-voltage direct current (VSC-HVDC) transmission lines. This method can locate faults with small training datasets. However, executing the task of transient FD for TCS in dynamic operation conditions is still an urgent problem that needs to be solved.

Motivated by the discussions above, we propose a TL strategy to detect the transient faults of TCS under various operation conditions. In the proposed method, a Cycle-Flow adversarial network (CFAN) is first constructed for latent feature extraction and data reconstruction in steady operation conditions. Secondly, a TL framework with the federated CFANs jointly adjust the changed information caused by varied operation conditions. The two mentioned steps are to learn and preserve knowledge under normal cases. Finally, designed federated CFANs reconstruct residuals with faulty data for transient FD under dynamic operation conditions.

The contributions of the proposed method are summarized as follows:A CFAN is proposed for latent variable extraction and data reconstruction, which consists of an invertible flow model and two discriminative networks; the loss function is designed as well. Specifically, bidirectional optimization can enhance the quality of reconstruction while mitigating interference caused by background noise through adversarial training and flexible inference.The proposed federated CFAN-based TL is divided into two stages. Initially, the first CFAN model is trained using normal data in steady operation conditions. Subsequently, the second CFAN calibrates the changed information caused by varied operation conditions utilizing limited data. In conclusion, the federated CFANs can jointly learn latent knowledge in a steady state and be applied to transient fault detection in various operation conditions.Simulation experiments are conducted on various transient faults using the normal steady state of TCS as the source domain and the dynamic operation condition as the target domain. The simulation results show that the federated CFAN-based TL method can improve the performance of transient fault detection.

The remainder of this paper is organized as follows: Section 2 states the transient fault detection problems and flow basics. Section 3 details the proposed transfer learning fault detection strategy based on federated CFANs. In Section 4, the experiment results and data sources are briefly described. Finally, the conclusions and prospects are given in Section 5.

## 2. Background and Preliminaries

### 2.1. Problem Statement

The schematic diagram of the TCS is shown in Figure 1. The pantograph delivers single-phase AC power from the public grid to the transformer. The rectifier receives a lower voltage $u_{n}$ and current $i_{n}$ from the transformer and converts single-phase AC into DC voltages ($u_{cd1}$, $u_{cd2}$) stabled by DC-link. The inverter then outputs three-phase AC voltage/current ($u_{u}/i_{sa}$, $u_{v}/i_{sb}$, $u_{w}/i_{sc}$) to drive the asynchronous traction motors. In addition, the traction control unit (TCU) receives the sensor signals and sends the gate control signals spwm and svpwm.

As the attended time of high-speed trains increases, irreversible scenarios will arise in components of the TCS [1]. TFs caused by these irreversible changes are temporary faults but may not necessarily cause permanent damage. TFs are usually induced by the internal structural defects and manufacturing processes of active components. Furthermore, noise signals such as electromagnetic interference, spark discharge, lightning strikes, load fluctuation, etc., also contribute to TFs.

There is analog signal interference in its external communication connections for TCU faults. Consider the three-phase current, $i_{sa,sb,sc}$, where its fault current is as follows:(1)$$i_{sa,sb,sc}^{f}=i_{sa,sb,sc}^{0}+f\left(p,q,A\right)$$
where f(p,q,A) represents transient pulses described by a double-exponential module, and p and q are the time coefficients of the injection signal, which codetermines the width of the injection pulse, rising time, and falling time. A is the amplitude coefficient of the injection signal. The control strategy will compensate for the aforementioned fault by leveraging closed-loop regulation, thereby rendering them challenging to detect or diagnose using conventional methodologies.

Furthermore, sensor faults caused by surges in power, ground wires, and grid-side voltage fluctuations are also causes of TFs. For the value of the U-phase current $i_{u}$, its fault current is as follows:(2)$$i_{u}^{f}\left(t\right)=i_{u}^{0}\left(t\right)+\delta \left(t\right)$$
where $i_{u}^{0}$ is the current value in a normal state, and δ(t) is the short-duration pulse value caused by the above factors.

In addition, soft errors of components in the traction control unit can also lead to non-permanent mutations in the output of the sensors. TFs usually appear randomly and disappear in a short period, which results in uncertainty [18].

### 2.2. Preliminaries of Normalizing Flow

Normalizing flow (NF) is a transformation of a simple probability distribution into a more complex distribution by a sequence of invertible and differentiable mappings, which allows for an exact likelihood calculation [28,29]. Therefore, NF has been widely used in image processing, denoising, and anomaly detection [30,31,32,33]. Suppose x is a high-dimensional random vector with a known probability density function (PDF) $p_{x}\left(x\right)$. The latent variable *z* is typically assumed to follow a specific distribution, usually the multivariate unit Gaussian distribution N(0, I), which compels the model to learn the input data distribution. Assuming $z\sim p_{z}\left(z\right)$, x and z are all D-dimensional; the PDFs of the given data are as follows:(3)$$\int_{z}p_{z}\left(z\right)dz=\int_{x}p_{x}\left(x\right)dx=1$$
(4)$$\left|p_{z}\left(z\right)\cdot dz\right|=\left|p_{x}\left(x\right)\cdot dx\right|$$
(5)$$p_{x}\left(x\right)=p_{z}\left(z\right)\left|\frac{dz}{dx}\right|=p_{z}\left(z\right)\left|\det\left(\frac{\partial z}{\partial x}\right)\right|$$
(6)$$\log p_{x}\left(x\right)=\log p_{z}\left(z\right)+\log\left|det(\frac{\partial z}{\partial x})\right|$$

The generation process can be expressed as follows:(7)$$z=f\left(x\right),\;x=g\left(z\right)$$
where f(·) is a reversible function that transforms a random variable x into z, which is also called bijection. g(·) is the inverse function of f(·) such that given for a data x, the variable inference is completed by $z=f\left(x\right)=g^{-1}(x)$, and θ is the parameter.

Therefore, (5) and (6) can be written as follows:(8)$$p_{x}\left(x\right)=p_{z}\left(z\right)\left|det\left(\frac{\partial f}{\partial x}\right)\right|{=p}_{z}\left(z\right)\left|\det J\left(Z\right)\right|$$
(9)$$\log p_{x}\left(x\right)=\log p_{z}\left(z\right)+\log\left|det\left(\frac{\partial f}{\partial x}\right)\right|=\log p_{z}\left(z\right)+\log\left|\det J\left(Z\right)\right|$$
where J(Z) is the B×B Jacobian matrix.

As shown in Figure 2, transformation g molds the PDF $p_{z}\left(z\right)$ into $p_{x}\left(x\right)$. The absolute Jacobian determinant $\left|\det J\left(Z\right)\right|$ quantifies the relative volume change in a small neighborhood around z due to g [34].

Based on the above, NF can complete the distribution transformation of any complexity.

As shown in (7). Considering the forward process, fitting a flow-based model f(·) can be achieved by minimizing the *Kullback*–*Leibler (KL*) divergence between the target distribution and the $p_{z}\left(z\right)$ can be expressed as follows:(10)$$\begin{array}{ll} L\left(\theta \right) & =D_{KL}[p_{x}^{*}\left(x\right)||p_{x}\left(x\right)]\\ & =-E_{p_{x}^{*}\left(x\right)}\left[\log p_{x}\left(x\right)\right]+c\\ & =-E_{p_{x}^{*}\left(x\right)}\left[\log p_{z}\left(z\right)+\log\left|\det J\left(Z\right)\right|\right]+c\\ & =-E_{p_{x}^{*}\left(x\right)}\left[\log p_{z}\left(g\left(z\right)\right)+\log\left|\det J\left(Z\right)\right|\right]+c \end{array}$$
(11)c=−M⋅log⁡a
where a is determined by the discretization level of the data and M is the dimension of z. Assuming the target data samples ${\left\{z(n)\right\}}_{n=1}^{N}$ from target distribution, the expectation of target distribution can be estimated by Monte Carlo as follows:(12)$$L\left(\theta \right)\approx -\frac{1}{N}\sum_{n=1}^{N}\log p_{z}\left(g(z(n)\right))+\log\left|\det J\left(z\left(n\right)\right)\right|+c$$

## 3. The Proposed Federated CFAN-Based Transfer Learning Strategy

Motivated by the research on image processing based on NF [35], a federated CFAN-based TL strategy to detect transient faults in TCSs is proposed due to its reversibility and flexibility in modeling various distributions.

In this work, the source domain is represented by $D^{s}$, where d represents data, which denotes the measurements under the steady operation of TCSs. Similarly, target domain data are represented by $D^{t}$, which represents the measurements in dynamic operation. There will be some differences in data distribution between the source and target domain. New knowledge can be acquired through reasonable adjustments of previous knowledge. This transfer-learning approach can achieve better FD performance than using only the target domain data [36].

### 3.1. Principle of CFAN

The framework of the proposed CFAN model is shown in Figure 3. In this work, the forward process of CFAN can be defined as H, and the reverse process is expressed as $H^{-1}$. Consider source domain sample ${\left\{d_{s}(k)\right\}}_{k=1}^{M}\in D^{s}$. The target of the defined model H is to learn the potential features of the source domain in which $\theta_{1}$ represents hyperparameters. Model H contains two mapping functions, the forward process H, and the reverse process $H^{-1}$. In addition, two adversarial discriminative networks, $D_{f}$ and $D_{r}$, are introduced, where $D_{f}$ aims to differentiate between $d_{s}(k)$ and the generated data $H(d_{s}\left(k\right);\theta_{1})$. Similarly, $D_{r}$ aims to distinguish between $d_{s}(k)$ and $H^{-1}(d_{s}\left(k\right);\theta_{1})$. $D_{f}$ encourages H to transform $d_{s}(k)$ into an output (itself) that is indistinguishable from $d_{s}(k)$ and vice versa for $D_{r}$ and $H^{-1}$.

Given a B dimensional input, $d_{s}(k)$:(13)$$d_{s}\left(k\right)=\left[\begin{array}{c} d_{s}^{1}\left(k\right)\\ \vdots \\ d_{s}^{B}\left(k\right) \end{array}\right]\in R^{B}$$

$d_{s}\left(k\right)$ is split into $d_{s}^{1:b}\left(k\right)$ and $d_{s}^{b+1:B}\left(k\right)$, which are given as follows:(14)$$d_{s}^{1:b}\left(k\right)=\left[\begin{array}{c} d_{s}^{1}\left(k\right)\\ \vdots \\ d_{s}^{b}\left(k\right) \end{array}\right]\in R^{b}$$
(15)$$d_{s}^{b+1:B}\left(k\right)=\left[\begin{array}{c} d_{s}^{b+1}\left(k\right)\\ \vdots \\ d_{s}^{B}\left(k\right) \end{array}\right]\in R^{B-b}$$

The affine coupling layer is presented in Figure 4; the output $h\left(k\right)$ of an affine coupling layer follows Equations (16) and (17).
(16)$$h_{1:b}\left(k\right)=d_{s}^{1:b}\left(k\right)$$
(17)$$h_{b+1:B}\left(k\right)=d_{s}^{b+1:B}\left(k\right)⨀exp(s(d_{s}^{1:b}\left(k\right)))+t\left(d_{s}^{1:b}\left(k\right)\right)$$

Finally, $h_{1:b}\left(k\right)$ and $h_{b+1:B}\left(k\right)$ are merged into one group $h\left(k\right)$.

As the reverse input $h\left(k\right)$ and output $x\left(k\right)$, its reverse process can be expressed as follows:(18)$$x_{1:b}\left(k\right)=h_{1:b}\left(k\right)=d_{s}^{1:b}\left(k\right)$$
(19)$$x_{b+1:B}\left(k\right)=\frac{h_{b+1:B}\left(k\right)-t\left(h_{1:b}\left(k\right)\right)}{exp(s(h_{1:b}\left(k\right)))}=\frac{d_{s}^{b+1:B}\left(k\right)-t\left(d_{s}^{1:b}\left(k\right)\right)}{exp(s(d_{s}^{1:b}\left(k\right)))}$$
where $s\left(\cdot \right)$ represents the scaling function, $t\left(\cdot \right)$ represents the translation function, and ⨀ is the Hadamard or element-wise product.

Considering the forward process, the Jacobian matrix of transformation f(·) can be expressed as follows:(20)$$\frac{\partial h}{\partial {d_{s}\left(k\right)}^{T}}=\left[\begin{array}{cc} identity & 0\\ \frac{\partial h_{b+1:B}}{\partial {d_{s}^{1:b}\left(k\right)}^{T}} & diagonal \end{array}\right]$$

The upper left area of the Jacobian matrix is an identity matrix I. Since $d_{s}^{1:b}\left(k\right)$ is irrelevant to $h_{b+1:B}\left(k\right)$, the upper right area of the Jacobian matrix is a zero matrix, 0. The lower right area of the Jacobian matrix is a diagonal matrix with the diagonal element $exp(s(d_{s}^{1:b}\left(k\right)))$. Therefore, the calculation of the lower left area of the Jacobian matrix can be ignored. Because the Jacobian of $s\left(\cdot \right)$ or $t\left(\cdot \right)$ is not necessary for computing the Jacobian determinant of the coupling layer, $s\left(\cdot \right)$ or $t\left(\cdot \right)$ can be arbitrarily complex for various network designs.

Although the coupling layer may be powerful, the distribution is often very complex in practice. Moreover, it is challenging to transform a complex distribution into another; one transformation is often insufficient. In addition, the forward transformation leaves some components unchanged, with the first *d* dimensions being identical to the initial data. Figure 5 illustrates the composition of the coupling layer in an alternating pattern. This structure allows different parts of the data to be passed through different transformation paths. It ensures that the final generated data do not contain components originating from the initial data [30]. Combining coupling layers is carried out as follows:(21)$$H=f_{1}◦f_{2}◦\cdots ◦f_{k}$$

Then, its reverse process can be expressed as follows:(22)$$H^{-1}={\left(f_{k}{◦\cdots ◦f_{2}◦f}_{1}\right)}^{-1}={f_{1}}^{-1}◦{f_{2}}^{-1}◦\cdots ◦{f_{k}}^{-1}$$

As mentioned above, to minimize the error between the input and reconstructed output, the expectation of $p_{D^{s}}(d_{s}(k))$ can be estimated by Monte Carlo as follows:(23)$$L\left(\theta \right)\approx -\frac{1}{N}\sum_{n=1}^{N}\log p_{d_{s}\left(k\right)\sim D^{s}}\left(H(d_{s}(k);\theta_{1}\right)))+\log\left|\det J_{H}\left(d_{s}\left(k\right)\right)\right|$$

Similarly, for the reverse process,
(24)$$L\left(\theta \right)\approx -\frac{1}{N}\sum_{n=1}^{N}\log p_{d_{s}\left(k\right)\sim D^{s}}\left(H^{-1}(d_{s}(k);\theta_{1}\right)))+\log\left|\det J_{H^{-1}}\left(d_{s}\left(k\right)\right)\right|$$

For the forward process, the loss function ${loss}_{H}$ of the model H in this work can be expressed as follows:(25)$${loss}_{H}=E_{d_{s}\left(k\right)\sim D^{s}}\left[\log D_{f}\left(d_{s}\left(k\right)\right)\right]+E_{d_{s}\left(k\right)\sim D^{s}}\left[\log\left(1-D_{f}\left(H\left(d_{s}\left(k\right);\theta_{1}\right)\right)\right)\right]$$
where $D_{f}$ is a discriminative network and $\theta_{f}$ is a hyperparameter, and then the loss function of $D_{f}$ can be expressed as follows:(26)$${loss}_{D_{f}}=E_{d_{s}\left(k\right)\sim D^{s}}\left[{\left(D_{f}\left(d_{s}\left(k\right)\right)-1\right)}^{2}\right]+E_{d_{s}\left(k\right)\sim D^{s}}\left[{\left(D_{f}\left(H\left(d_{s}\left(k\right);\theta_{1}\right)\right)\right)}^{2}\right]$$

Similarly, the loss function ${loss}_{H^{-1}}$ of the reverse process can be expressed as follows:(27)$${loss}_{H^{-1}}=E_{d_{s}\left(k\right)\sim D^{s}}\left[\log D_{r}\left(d_{s}\left(k\right)\right)\right]+E_{d_{s}\left(k\right)\sim D^{s}}\left[\log\left(1-D_{r}\left(H^{-1}\left(d_{s}\left(k\right);\theta_{1}\right)\right)\right)\right]$$
where $D_{r}$ is a discriminative network, and $\theta_{r}$ is a hyperparameter, and then the loss function of $D_{r}$ can be expressed as follows:(28)$${loss}_{D_{r}}=E_{d_{s}\left(k\right)\sim D^{s}}\left[{\left(D_{r}\left(d_{s}\left(k\right)\right)-1\right)}^{2}\right]+E_{d_{s}\left(k\right)\sim D^{s}}\left[{\left(D_{r}\left(H^{-1}\left(d_{s}\left(k\right);\theta_{1}\right)\right)\right)}^{2}\right]$$

The total loss $L_{total}$ of the proposed CFAN is presented as follows:(29)$$L_{total}={loss}_{H}+{loss}_{H^{-1}}+{loss}_{D_{f}}+{loss}_{D_{r}}$$

The overall optimization objective of the model can be written as follows:(30)$$\langle \theta_{1}^{*},\theta_{f}^{*},\theta_{r}^{*}\rangle =arg\underset{H,H^{-1}}{\min}\underset{D_{f},D_{r}}{\max}L_{total}$$

In summary, the CFAN can learn knowledge in the source domains by adversarial training, and the trained hyperparameter is ${\theta_{1}}^{*}$.

The trained ${CFAN}^{*}$ model can perform FD tasks under steady operation conditions(The training progress is detailed in Algorithm 1). However, the distributed discrepancies arising from diverse operational conditions result in a decline in its overall performance. To mitigate this issue, fine-tuning of the model is necessary to attain optimal FD performance through TL.
**Algorithm 1:** offlineLoop **for** number of training iterations do $\mathrm{Sample}\;\mathrm{from}\;\mathrm{dataset}\;D^{s}$$,\;{\left\{d_{s}(k)\right\}}_{k=1}^{M}\epsilon D^{s}$
 Learning H, $H^{-1}$: $\mathrm{For}\;d_{s}(1)$$,\;\mathrm{compute}\;H(d_{s}\left(1\right);\theta_{1})$$,H^{-1}(d_{s}\left(1\right);\theta_{1})$
 $\mathrm{Backward}\;\mathrm{propagate}\;{loss}_{H}$,${loss}_{H^{-1}}$, $\mathrm{update}\;\theta_{1}$ by Adam optimizer [36] $\mathrm{Learning}\;D_{f}$, $D_{r}$: $\mathrm{For}\;d_{s}(1)$$,\;\mathrm{compute}\;{D_{f}(d}_{s}(1))$$,\;{D_{r}(d}_{s}(1))$
 $\mathrm{Backward}\;\mathrm{propagate}\;{loss}_{D_{f}}$$,\;{loss}_{D_{r}}$, $\mathrm{update}\;\theta_{f}$, $\theta_{r}$ by Adam optimizer $\mathrm{For}\;d_{s}(1)$$,\;\mathrm{compute}\;D_{f}\left(H\left(d_{s}(1)\right)\right)$, $D_{r}\left(H^{-1}\left(d_{s}(1)\right)\right)$
 $\mathrm{Backward}\;\mathrm{propagate}\;{loss}_{D_{f}}$, ${loss}_{D_{r}}$, $\mathrm{update}\;\theta_{f}$, $\theta_{r}$ by Adam optimizerend forend loop

### 3.2. Fault Detection with Transfer Learning Based on Federated CFANs

This work aims to establish an FD model under dynamic operation with TL. The first CFAN reflects the information on steady-state operation in the system, which was trained in the previous step. The second CFAN learns the performance changes influenced by domain changes. This design concept involves neural model-aided learning to identify changing and unchanging crucial parameters. The framework of the proposed TL strategy is illustrated in Figure 6.

The data of target domain $D^{t}$ are input into the ${CFAN}_{1}^{*}$ after training. Due to the different data distribution between the $D^{s}$ and $D^{t}$, their performance will also change. Consider target domain sample ${\left\{d_{t}(n)\right\}}_{n=1}^{L}$ from $D^{t}$, where the residual signal $e_{1}(n)\sim E_{1}$ can be expressed as follows:(31)$$e_{1}\left(n\right)=d_{t}\left(n\right)-{CFAN}_{1}^{*}\left(d_{t}\left(n\right);{\theta_{1}}^{*}\right)$$

From the above formula, $e_{1}$ is the path between the source and the target domain, which retains the information when the operation conditions change. The ${CFAN}_{2}$ has the ability to calibrate the knowledge changes caused by the varied operation conditions. The construction of ${CFAN}_{2}$ is similar to ${CFAN}_{1}$, and $\theta_{2}$ is a hyperparameter. In addition, it also includes the discriminative network $D_{f2}$ and $D_{r2}$, in which $\theta_{f2}$ and $\theta_{r2}$ are hyperparameters. The loss function ${loss}_{H_{2}}$ of ${CFAN}_{2}$ can be expressed as follows:(32)$${loss}_{H_{2}}=E_{d_{t}\left(n\right)\sim D^{t}}\left[\log D_{f2}\left(e_{1}\left(n\right)\right)\right]+E_{d_{t}\left(n\right)\sim D^{t}}\left[\log\left(1-D_{f2}\left(H_{2}\left(d_{t}\left(n\right);\theta_{2}\right)\right)\right)\right]$$

The loss function ${loss}_{D_{f2}}$ of $D_{f2}$ can be expressed as follows:(33)$${loss}_{D_{f2}}=E_{d_{t}\left(n\right)\sim D^{t}}\left[{\left(D_{f2}\left(e_{1}\left(n\right)\right)-1\right)}^{2}\right]+E_{d_{t}\left(n\right)\sim D^{t}}\left[{\left(D_{f2}\left(H_{2}\left(d_{t}\left(n\right);\theta_{2}\right)\right)\right)}^{2}\right]$$

The loss function of the reverse process $H_{2}^{-1}$ can be expressed as follows:(34)$${loss}_{H_{2}^{-1}}=E_{d_{t}\left(n\right)\sim D^{t}}\left[\log D_{r2}\left(d_{t}\left(n\right)\right)\right]\;\;\;\;\;\;\;\;\;\;\;+E_{e_{1}\left(n\right)\sim E_{1}}\left[\log\left(1-D_{r2}\left(H_{2}^{-1}\left(e_{1}\left(n\right);\theta_{2}\right)\right)\right)\right]$$

The loss function ${loss}_{D_{r2}}$ of $D_{r2}$ can be expressed as follows:(35)$${loss}_{D_{r2}}=E_{d_{t}\left(n\right)\sim D^{t}}\left[{\left(D_{r2}\left(d_{t}\left(n\right)\right)-1\right)}^{2}\right]+E_{e_{1}\left(n\right)\sim E_{1}}\left[{\left(D_{r2}\left(H_{2}^{-1}\left(e_{1}\left(n\right);\theta_{2}\right)\right)\right)}^{2}\right]$$

In summary, the total loss $L_{total2}$ of ${CFAN}_{2}$ is formulated as follows:(36)$$L_{total2}={loss}_{H_{2}}+{loss}_{H_{2}^{-1}}+{loss}_{D_{f2}}+{loss}_{D_{r2}}$$

The overall optimization objective of the proposed TL model is provided as follows:(37)$$\langle \theta_{2}^{*},\theta_{f2}^{*},\theta_{r2}^{*}\rangle =arg\underset{H_{2},H_{2}^{-1}}{\min}\underset{D_{f2},D_{r2}}{\max}L_{total2}$$

The ${CFAN}_{2}^{*}$ learns the performance variation of the ${CFAN}_{1}^{*}$ due to varied operation conditions; the training process of ${CFAN}_{2}^{*}$ is detailed in Algorithm 2. The change information ${\hat{e}}_{1}\left(k\right)$ is obtained using the following formula:(38)$${\hat{e}}_{1}\left(m\right)={CFAN}_{2}^{*}\left(d_{t}\left(m\right);{\theta_{2}}^{*}\right)$$
**Algorithm 2:** offlineLoop **for** number of training iterations do $\mathrm{Sample}\;\mathrm{from}\;\mathrm{dataset}\;{\left\{d_{t}(n)\right\}}_{n=1}^{L}\epsilon D^{t}$
 $\mathrm{Learning}\;H_{2}$,$H_{2}^{-1}$: $\mathrm{For}\;d_{t}(1)$$,\;\mathrm{compute}\;e_{1}\left(n\right)=d_{t}\left(1\right)-{CFAN}_{1}^{*}(d_{t}(1);{\theta_{1}}^{*})$
 $\mathrm{For}\;d_{t}(1)$$,\;\mathrm{compute}\;H_{2}(d_{t}(1);\theta_{2})$$,\;aH_{2}^{-1}(e_{1}\left(n\right);\theta_{2})$
 $\mathrm{Backward}\;\mathrm{propagate}\;{loss}_{H_{2}}$$,\;{loss}_{H_{2}^{-1}}$$,\;\mathrm{update}\;\theta_{2}$ by Adam optimizer $\mathrm{Learning}\;D_{f2}$$,\;D_{r2}$: $\mathrm{For}\;e_{1}\left(n\right)$$,\;d_{t}\left(1\right)$$,\;\mathrm{compute}\;D_{f2}$$(e_{1}\left(n\right)$$),\;D_{r2}$$(d_{t}\left(1\right)$) $\mathrm{Backward}\;\mathrm{propagate}\;{loss}_{D_{f2}}$$,\;{loss}_{D_{f2}}$, $\mathrm{update}\;\theta_{f2}$, $\theta_{r2}$ by Adam optimizer $\mathrm{For}\;d_{t}\left(1\right)$$,\;e_{1}\left(n\right)$$,\;\mathrm{compute}\;D_{f2}\left(H_{2}(d_{t}(1);\theta_{2})\right)$, $D_{r2}\left(H_{2}^{-1}(e_{1}\left(n\right);\theta_{2}\right)$
 $\mathrm{Backward}\;\mathrm{propagate}\;{loss}_{D_{f2}}$, ${loss}_{D_{r2}}$, $\mathrm{update}\;\theta_{f2}$, $\theta_{r2}$ by Adam optimizerend forend loop

Based on the above analysis, the residual signal r(m) used for the final FD decision is defined as follows:(39)$$r\left(m\right)={e_{1}\left(m\right)-\hat{e}}_{1}\left(m\right)=d_{t}\left(m\right)-{CFAN}_{1}^{*}\left(d_{t}\left(m\right);{\theta_{1}}^{*}\right)-{CFAN}_{2}^{*}\left(d_{t}\left(m\right);{\theta_{2}}^{*}\right)$$

According to the final decision signal r, m represents the dimension of r. The framework of the proposed federated CFANs is depicted in Figure 7.

This work utilizes the root mean square (RMS) norm to maintain satisfactory false alarm rates (FARs) in high-dimensional situations. The RMS measures the average energy of a signal r and is defined by the following formula:(40)$${J(r(m))}_{RMS}=\frac{1}{n}{\left(r(m\right)}^{T}r(m))$$

The threshold is set to be
(41)$$J_{th}=\underset{\mathrm{fault}-\mathrm{free}}{\sup}{J\left(r\right)}_{RMS}$$

Then, the fault detection logic becomes
(42)$$\left\{\begin{array}{c} {J\left(r\left(m\right)\right)}_{RMS}\leq J_{th}⟹no\;alarm,\;fault-free\\ {J\left(r\left(m\right)\right)}_{RMS}>J_{th}⟹alarm,\;a\;fault\;is\;detected. \end{array}\right.$$

The flowchart of the proposed method is illustrated in Figure 8, comprising an offline training phase and an online fault detection (FD) phase. The first CFAN-based model ${CFAN}_{1}$ is trained by using the normal data $D^{s}$ obtained during steady operation conditions to extract latent variables and reconstruct data. Subsequently, the model ${CFAN}_{2}$ undergoes federated training based on dynamic operation condition data $D^{t}$. The trained federated neural networks ${CFAN}_{1}^{*}$ and ${CFAN}_{2}^{*}$ enable feature extraction and the reconstruction of the healthy data. Thus, the residual r is calculated using the federated CFANs. Finally, with the FD threshold $J_{th}$ being determined by the RMS statistics of the residual r, the ${J\left(r\left(m\right)\right)}_{RMS}$ of the testing data is compared with $J_{th}$ to realize the FD of the TCS.

## 4. Experiment Results and Analysis

In this section, the data source and experimental platform are briefly described. To verify the effectiveness of the proposed method, FD tasks with different methods were performed on the TCS under dynamic operation conditions. Some discussions are proposed based on the experimental results.

### 4.1. Data Description

In this case, a TCS is adopted to demonstrate the effectiveness of the proposed FD method. A simulation platform of traction drive control systems named “TDCS-FIB” is presented in [37,38]. TDCS-FIB develops fault injection benchmarks based on simulation models. TDCS-FIB provides a variety of fault injection types for the main components in TCS, which provides reliable data support for fault detection and diagnosis.

To verify the proposed method, a TCS with different TFs is adopted. As depicted in Figure 9, the onboard TCS serves as the experimental system, with its specifications presented in Table 1. The sensor data were collected under traction operation conditions.

In practice, transient faults will lead to abnormal data from multiple sensors. Multi-sensor FD can reduce interference and improve detection efficiency [39,40]. Therefore, multi-sensor data are used to detect transient faults, which include the three-phase current output $\left[i_{sa}\;i_{sb}\;i_{sc}\right]$ of an inverter, the voltage output $\left[u_{cd1}\;u_{cd2}\right]$ of the upper and lower support capacitors in the DC link, and the transformer secondary voltage and current $\left[u_{n}\;i_{n}\right]$. The FD model of TCS is trained based on the sensor signals as follows:(43)$$\left[i_{sa}\;i_{sb}\;i_{sc}\;u_{cd1}\;u_{cd2}\;u_{n}\;i_{n}\right]\in D$$
where $\left[i_{sa}\;i_{sb}\;i_{sc}\;u_{cd1}\;u_{cd2}\;u_{n}\;i_{n}\right]\in D^{s},{\;D}^{t}$. The collected data can be expressed as $D^{f}$ for the transient faults under dynamic operation conditions.

Since the waveforms of the seven groups of sensors tend to be stable after the $1\times 10^{4}$-th step, $1\times 10^{3}$ samples in the normal steady state of the TCS are obtained as the source domain training dataset $D^{s}$, and $2\times 10^{2}$ samples in the dynamic condition and 50 in steady are used as the target domain training dataset $D^{t}$.

The test dataset in the dynamic state contains four transient faults and fault-free scenarios. Each fault scenario contains $5\times 10^{2}$ samples, and the fault-free scenario contains $1\times 10^{5}$ samples. The evaluation of the experimental results is completed using the false alarm rate, fault detection rate (FDR), recall, and accuracy rate (ACR), which are defined as follows:(44)$$FDR=\frac{{TP}_{Fx}}{{TP}_{Fx}+{FP}_{Fx}}$$
(45)$$FAR=\frac{FN}{TN+FN}$$
(46)$$recall=\frac{TP}{TP+FP}$$
(47)$$ACR=\frac{TP+TN}{TP+FP+TN+FN}$$

Define fault samples as positive samples and normal samples as negative samples. The total number of fault samples predicted to be correct is called true positive (TP). The total number of fault samples predicted to be errors is called false positive (FP). The total number of normal samples predicted to be correct is true negative (TN), and the total number of errors is false negative (FN). Fx represents the x type of fault.

The proposed model was built by Pytorch 1.13.1. The ${CFAN}_{1}$ and ${CFAN}_{2}$ models have the same structure and contain four affine coupling layers. $s\left(\cdot \right)$ includes two fully connected layers with 2×100 neurons. $t\left(\cdot \right)$ includes two fully connected layers with 2×100 neurons. The two discriminators $D\left(\cdot \right)$ use the same fully connected structure with (200,100,50,1). According to the loss function $L_{total}$ defined in (29) and $L_{total2}$ defined in (35), the best weights and biases can be obtained via ADAM. The details of the CFANs and methods for comparison are given in Table 2 and Table 3.

### 4.2. Analysis and Discussion

Comparisons between each FD task and other methods were conducted, encompassing four types of FD tasks and fault-free detection tasks for each method. Figure 10, Figure 11, Figure 12 and Figure 13 illustrate the FD results obtained using both the proposed method and VAE (including transfer and non-transfer learning). The traditional VAE refers to the VAE method without TL, while the federated VAE, which incorporates a similar TL strategy as our proposed method, is adapted for dynamic operating conditions. As shown in Figure 10a, Figure 11a, Figure 12a and Figure 13a, the blue curve represents the a-phase current waveform $i_{sa}$, and the orange dotted line represents the fault injection time. For (b), (c), and (d) in Figure 10, Figure 11, Figure 12 and Figure 13, the blue curve represents the detection results using three methods, and the red dotted line in the figures represents the FD threshold $J_{th}$.

The fault $F_{1}$ is attributed to the damage incurred by manufacturing processes, overstress, and other contributing factors on the shielding layer of communication cables. The transmission of external pulses in combinational logic circuits induces variations in both the pulse width and amplitude, which leads to TF in the TCS.

The reason for $F_{2}$ faults is that the sensor chip pins and wiring are loose or improperly connected. The sensor signal is instantaneously disturbed by vibration, thereby inducing transient fault $F_{2}$.

Transient shock faults $F_{3}$ may arise from improper sensor installation and the degradation of insulating materials triggered by power and ground wire surges.

The occurrence of $F_{4}$ can be attributed to IGBT damage resulting from internal structural defects, manufacturing processes, and other contributing factors. Furthermore, excessive stress induced by high temperatures may lead to gate driver circuit failure, such as TF caused by erroneous pulse control signals originating from the control circuit.

The comparison results of the three methods are illustrated in Figure 14, and Table 4 shows the ACR and average fault detection delay. The proposed method comprehensively achieves better performance for different FD tasks. Specifically, Figure 14a shows the FDR, and Figure 14b shows the FAR of different methods under four types of faults. The FDR of the other two FD methods is lower than that of the method described in this article. In Figure 14b and Table 3, the FAR, recall, and ACR of different methods are all lower than those of the method proposed in this work. The traditional VAE does not include the TL process and cannot adaptively adjust the changing knowledge based on the target domain data, which causes poor FD performance.

The data distributions vary across different operation conditions of TCSs, leading to a degradation in FD performance. However, there exists common knowledge among various operation conditions, necessitating the acquisition of knowledge from the steady-state operation of a TCS. As depicted in Figure 10, Figure 11, Figure 12 and Figure 13, due to the proposed TL strategy that leverages prior knowledge and mitigates the impact of operational variations, a federated VAE outperforms a traditional VAE. The proposed TL strategy based on federated CFANs effectively transfers and adapts knowledge between steady-state and dynamic operation conditions while ensuring the accurate extraction of latent variables and data reconstruction. By leveraging the adversarial training and reversibility properties of CFANs, the precise description of data distribution is achieved through bidirectional optimization, resulting in significant performance improvements as demonstrated in Figure 14 and Table 4. Especially for weak TFs (case studies $F_{1}$, $F_{2}$, and $F_{4}$), this proposed method exhibits superior fault detection capabilities under dynamic operating conditions.

In addition, FD experiments are also introduced under steady operating conditions. The test dataset in the steady state also contains four transient faults and fault-free scenarios which are similar to $F_{1}$, $F_{2}$,$\;F_{3}$, and $F_{4}$ in dynamic operation conditions. The performance comparison of different methods is shown in Table 5, each fault scenario contains $1\times 10^{3}$ samples, and the fault-free scenario contains $1\times 10^{5}$ samples. The comparison results of the three methods are illustrated in Figure 15 and Table 6 and Table 7.

The comparison results in steady operation conditions are illustrated in Figure 15, and Table 7 shows the FAR, recall, ACR, and average FD delay. It can be concluded that the proposed second CFAN achieves better performance for different FD tasks under steady operation conditions, for the reason that the knowledge of steady states has been learned by a small amount of data in healthy condition.

The training loss curve is defined by the Mean Squared Error (MSE) for evaluating the reconstruction accuracy. As illustrated in Figure 16a, during the training of the proposed method, the training loss stabilizes at a lower level than the other two methods, indicating the superior data reconstruction capabilities of the proposed method. Figure 16b displays the loss of ${CFAN}_{2}$ and federated VAE network2. The losses of two methods converge to a similar value, which illustrates that both networks have the ability to achieve performance adjustments.

The ROC-AUC (Receiver Operating Characteristic-Area Under the Curve) curves of three methods are shown in Figure 17, the AUC of the proposed method is 0.953, while the AUC values of the traditional VAE and the federated VAE are 0.826 and 0.907, respectively. The proposed method has the largest area under the curve, indicating superior performance in terms of FD.

Generally, to ensure the security of the system, the TCS typically works in normal states. As a result, the fault occurrences have a much lower chance of appearing than the healthy instances [21]. This unsupervised method only learns normal patterns from fault-free data, which is a feasible solution to the problem of imbalanced data. Therefore, unsupervised learning improves robustness without the cost of labeling. This FD method is not limited to the TCS of the train, but for faults in other electrical systems, this method has efficient transient FD performance.

## 5. Conclusions

In this work, we present a transient fault detection method under dynamic operation conditions. For the purpose of latent variable extraction and data reconstruction, a CFAN is established by an invertible flow model and two discriminative networks; additionally, the loss function was designed. Moreover, adversarial training and bidirectional optimization can enhance the reconstruction quality and depress interference caused by background noise.

Then, an unsupervised transfer learning strategy based on federated CFANs is proposed for transient fault detection under various operation conditions, which is divided into two stages. Initially, the first CFAN model is trained using the normal data in steady operation conditions. Subsequently, the second CFAN calibrates the changed information caused by varied operation conditions utilizing only a few samples. The federated CFANs can jointly learn latent knowledge in steady states and be applied to transient fault detection in various operation conditions.

By selecting the data-driven fault detection methods for comparative experiments, the effectiveness of the method is verified.

Several directions are available for future work. The first is to develop fault diagnosis technology and locate faulty components further. Otherwise, the FD method employed in this work is based on the CRH2 type, and data related to high-speed trains with different topological structures have not been explored. Such out-of-distribution (OOD) data, as mentioned in [41], may negatively impact FD performance. Future work will be considered, and fault diagnosis methods for high-speed trains of multiple types will be developed.

## Figures and Tables

**Figure 1 sensors-24-04839-f001:**
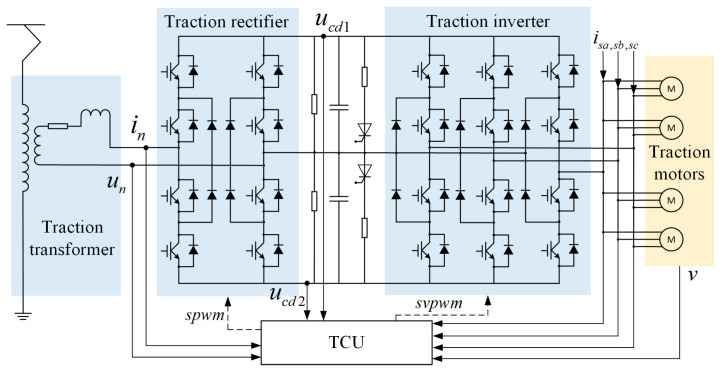
Circuit topology of TCS in high-speed trains.

**Figure 2 sensors-24-04839-f002:**
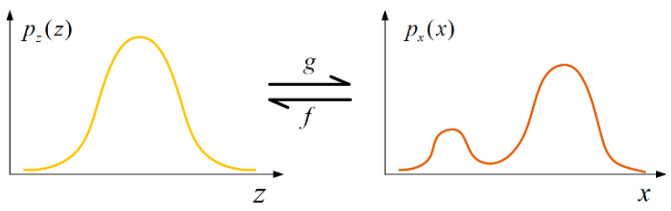
Change in variables.

**Figure 3 sensors-24-04839-f003:**
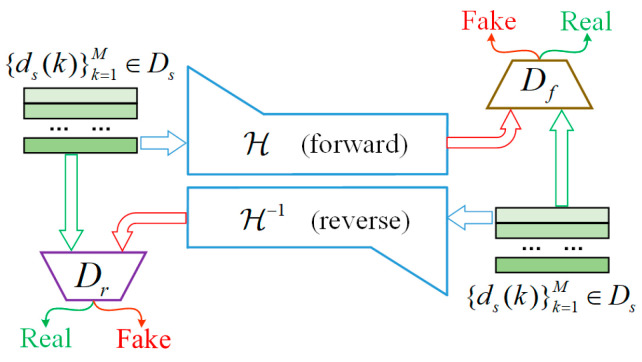
The framework of the CFAN model.

**Figure 4 sensors-24-04839-f004:**
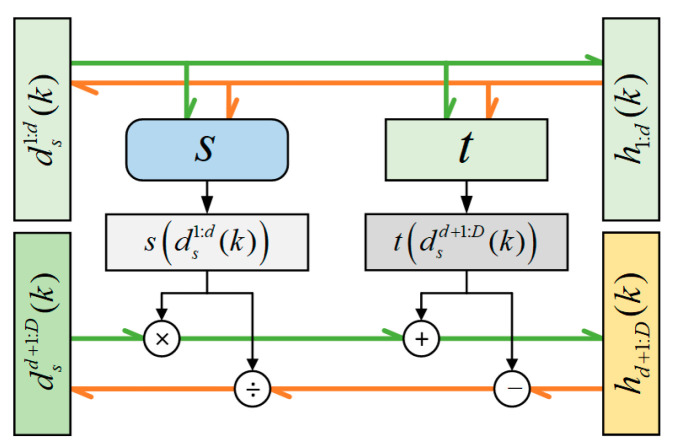
The structure of a coupling layer.

**Figure 5 sensors-24-04839-f005:**
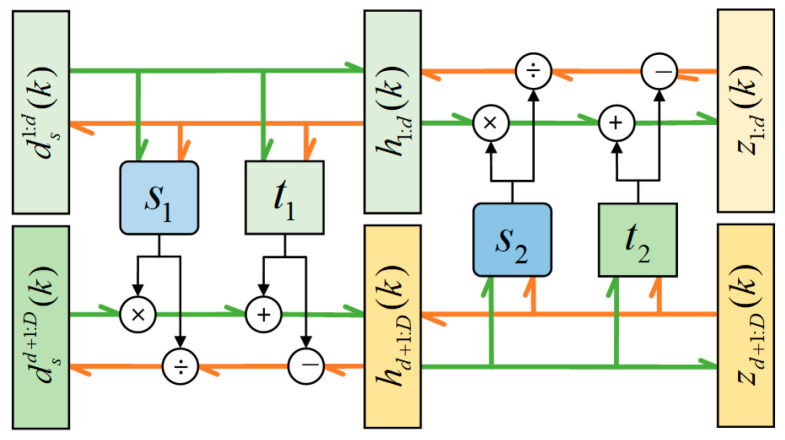
Coupling layer stacking.

**Figure 6 sensors-24-04839-f006:**
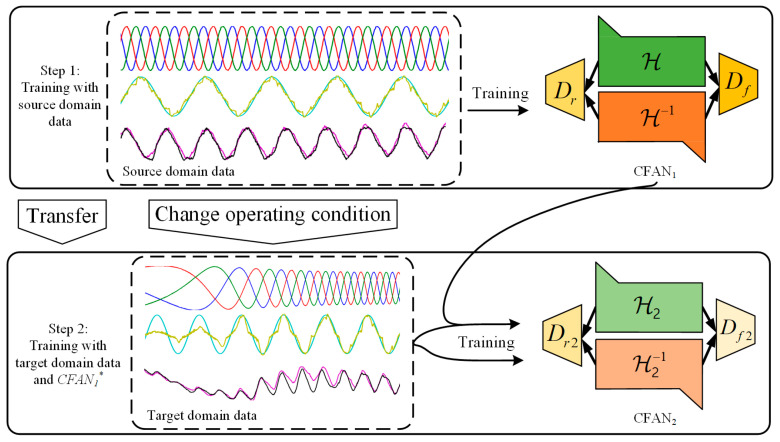
The federated CFAN-based TL strategy.

**Figure 7 sensors-24-04839-f007:**
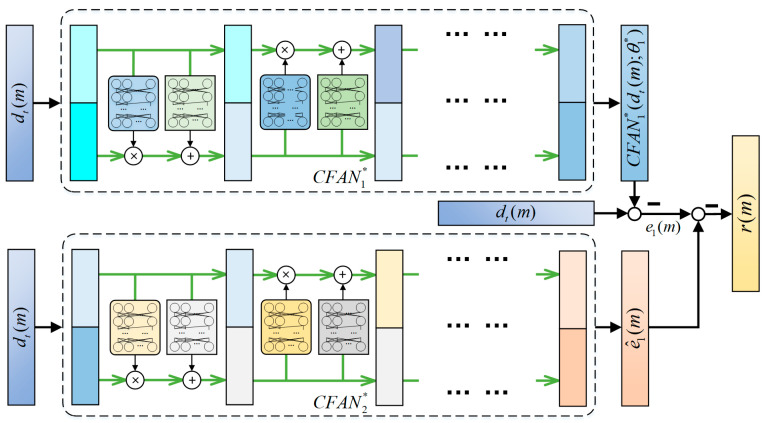
The structure of federated CFAN-based transfer learning.

**Figure 8 sensors-24-04839-f008:**
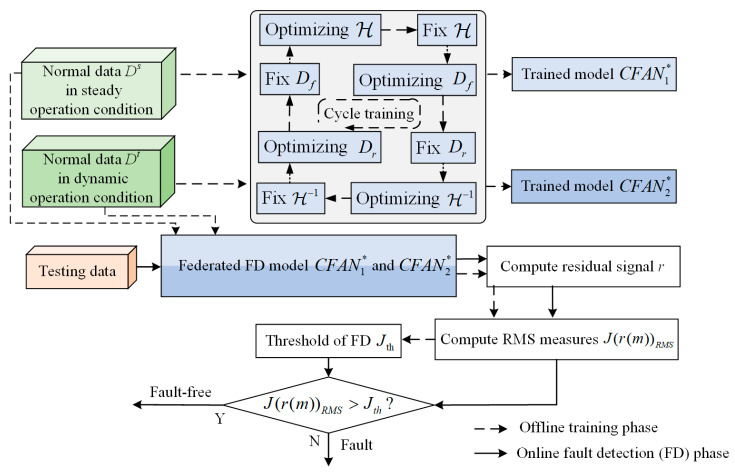
The overall flowchart of the proposed FD method.

**Figure 9 sensors-24-04839-f009:**
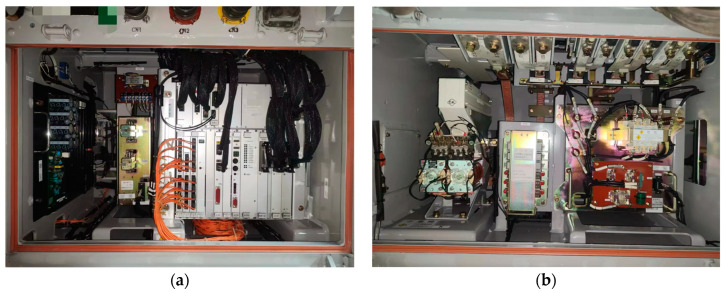
The onboard TCS in high-speed trains. (**a**) Traction control unit. (**b**) Main circuit of TCS.

**Figure 10 sensors-24-04839-f010:**
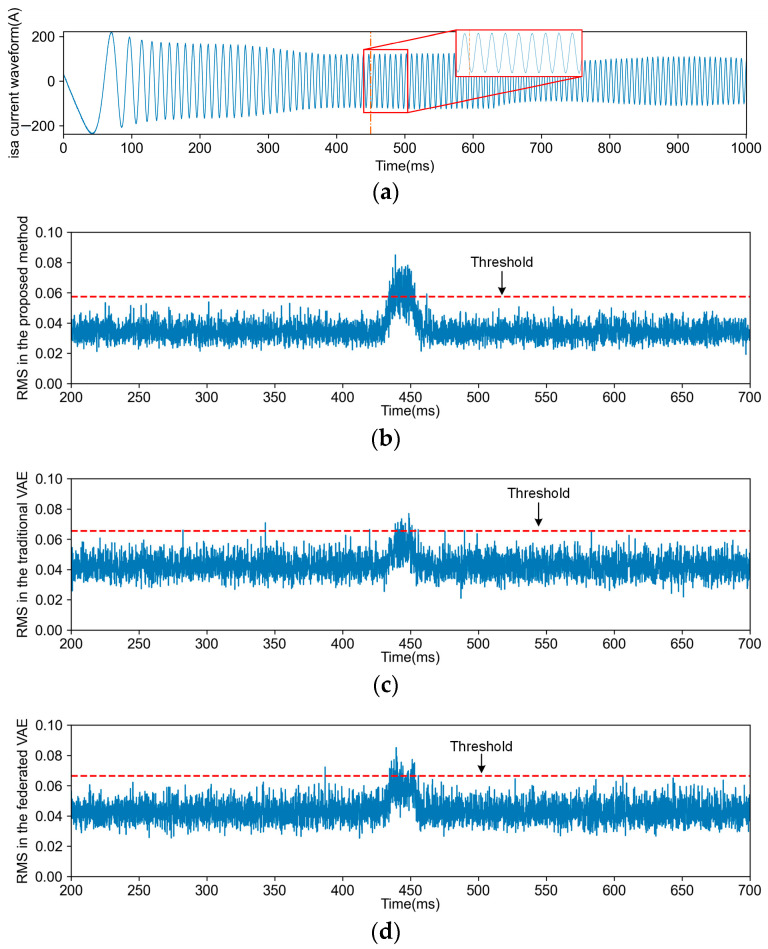
The current waveform and FD results for $F_{1}$. (**a**) is the traction motor a-phase current waveform $i_{sa}$ of the $F_{1}$ fault; (**b**) is the FD result obtained through the proposed method; (**c**) is the FD result obtained through the traditional VAE; (**d**) is the FD result obtained through the federated VAE.

**Figure 11 sensors-24-04839-f011:**
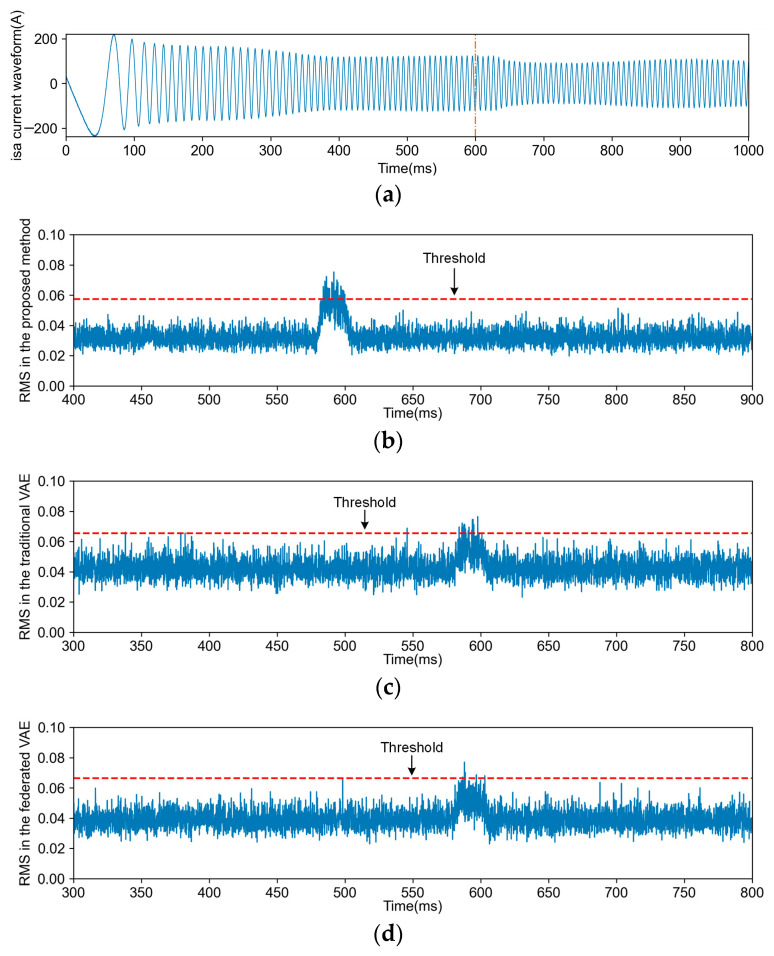
The current waveform and FD results for $F_{2}$. (**a**) is the traction motor a-phase current waveform $i_{sa}$ of the $F_{2}$ fault; (**b**) is the FD result obtained through the proposed method; (**c**) is the FD result obtained through the traditional VAE; (**d**) is the FD result obtained through the federated VAE.

**Figure 12 sensors-24-04839-f012:**
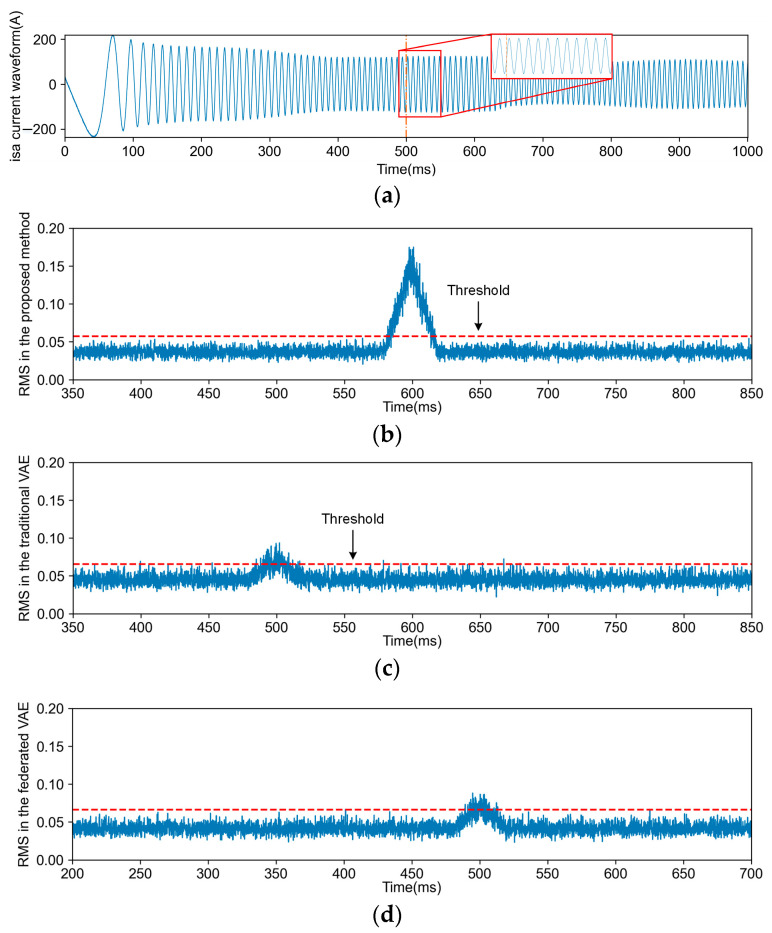
The current waveform and FD results for $F_{3}$. (**a**) is the traction motor a-phase current waveform $i_{sa}$ of the $F_{3}$ fault; (**b**) is the FD result obtained through the proposed method; (**c**) is the FD result obtained through the traditional VAE; (**d**) is the FD result obtained through the federated VAE.

**Figure 13 sensors-24-04839-f013:**
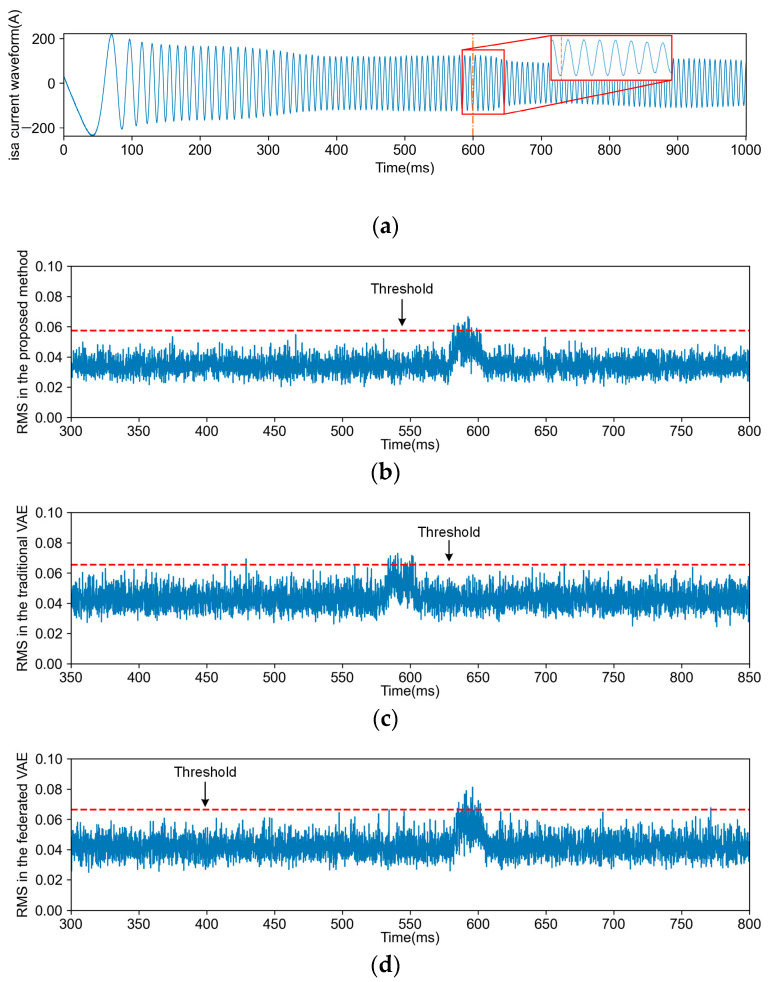
The current waveform and FD results for $F_{4}$. (**a**) is the traction motor a-phase current waveform $i_{sa}$ of the $F_{4}$ fault; (**b**) is the FD result obtained through the proposed method; (**c**) is the FD result obtained through the traditional VAE; (**d**) is the FD result obtained through the federated VAE.

**Figure 14 sensors-24-04839-f014:**
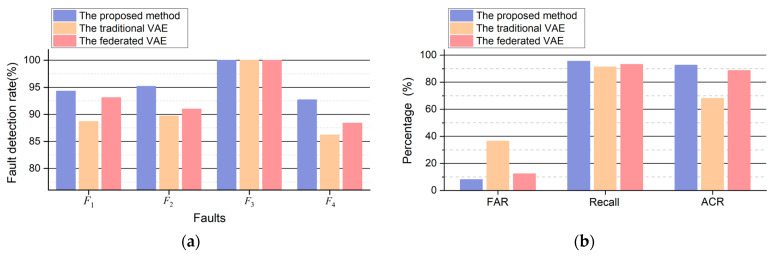
The comparison of results among different methods: (**a**) the FDR of four types of transient faults; (**b**) the FAR, recall, and ACR of three methods.

**Figure 15 sensors-24-04839-f015:**
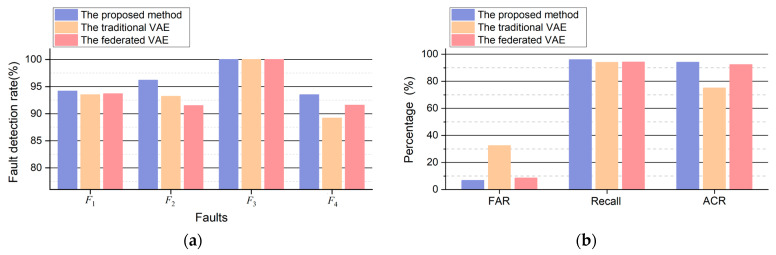
The comparison results among different methods in steady operation conditions: (**a**) the FDRs of three methods; (**b**) the FAR, recall, and ACR of three methods.

**Figure 16 sensors-24-04839-f016:**
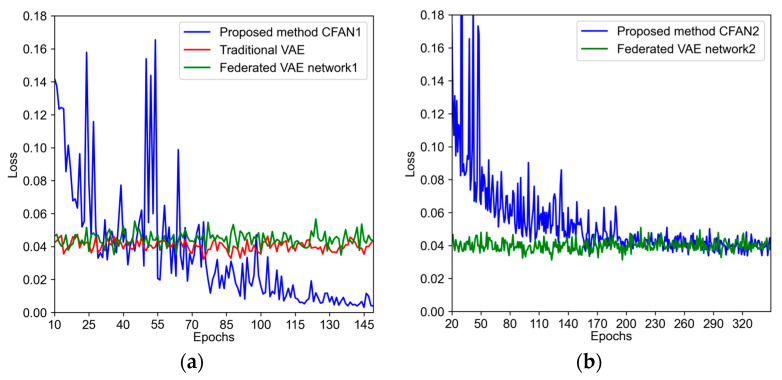
Loss of three methods. (**a**) Comparison of three methods for first network loss; (**b**) second network loss comparison of proposed method and federated VAE.

**Figure 17 sensors-24-04839-f017:**
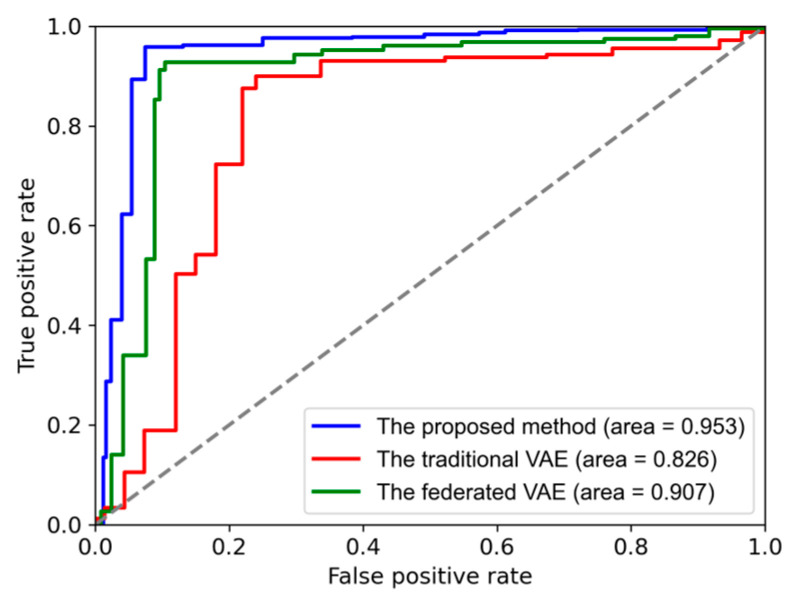
ROC-AUC of three methods.

**Table 1 sensors-24-04839-t001:** Specifications of experimental system under normal case.

Parameter Setting	Parameter Description	Value
$R_{s}$	Stator’s resistance	0.114 Ω
$R_{r}$	Rotor’s resistance	0.146 Ω
$L_{m}$	Magnetizing inductance	32.747 H
$P_{r}$	Rated power of traction motor.	300 KW
$n_{p}$	Motor pole pairs	2
$u_{d}$	Voltage of dc link	1500 V, 2600 V
$R_{n}$	Leakage resistance on line side	0.2 Ω
$L_{n}$	Leakage inductance on line side	0.002 H
$C_{d1},C_{d2}$	The filter capacitors of dc link	0.016 F
$R_{d1},R_{d2}$	The filter resistances of dc link	6000 Ω
$i_{u,v,w}$	Three-phase currents	±103 A
$V_{max}$	Speed	196 km/h

**Table 2 sensors-24-04839-t002:** Configuration of federated CFAN models.

Parameter Setting	$\mathrm{Structure}\;\mathrm{of}\;s\left(\cdot \right)$	$\mathrm{Structure}\;\mathrm{of}\;t\left(\cdot \right)$	$\mathrm{Structure}\;\mathrm{of}\;D\left(\cdot \right)$
$\mathrm{The}\;{CFAN}_{1}$ model	(100,100)	(100,100)	(200,100,50,1)
$\mathrm{The}\;{CFAN}_{2}$ model	(100,100)	(100,100)	(200,100,50,1)
Initial Learning rates	0.01	0.01	0.01
Activation functions	Relu, Sigmoid	Relu, Sigmoid	Relu, Relu, Relu, Sigmoid

**Table 3 sensors-24-04839-t003:** Configuration of methods for comparison.

Models for Comparison	Parameter Setting	Structure of Encoder	Structure of Decoder
The traditional VAE	Model	(200,50,10)	(10,50,200)
	Initial learning rates	0.01	0.01
	Activation functions	Relu	Relu
The federated VAE	First model	(200,50,10)	(10,50,200)
	Second model	(200,50,10)	(10,50,200)
	Initial learning rates	0.01	0.01
	Activation functions	Relu	Relu

**Table 4 sensors-24-04839-t004:** Detection results of different methods.

Faults	Methods	FDR
$F_{1}$	The proposed method	94.3%
	The traditional VAE	88.7%
	The federated VAE	93.1%
$F_{2}$	The proposed method	95.2%
	The traditional VAE	89.7%
	The federated VAE	91.0%
$F_{3}$	The proposed method	100%
	The traditional VAE	100%
	The federated VAE	100%
$F_{4}$	The proposed method	92.7%
	The traditional VAE	86.2%
	The federated VAE	88.4%

**Table 5 sensors-24-04839-t005:** Performance comparison of different methods.

Methods	FAR	Recall	ACR	Average FD Delay
The proposed method	8.1%	95.5%	92.5%	0.00133 s
The traditional VAE	36.5%	91.2%	68.1%	0.00142 s
The federated VAE	12.3%	93.1%	88.6%	0.00136 s

**Table 6 sensors-24-04839-t006:** Detection results of different methods in steady operation conditions.

Faults	Methods	FDR
$F_{1}$	The proposed method	94.2%
	The traditional VAE	93.5%
	The federated VAE	93.7%
$F_{2}$	The proposed method	96.2%
	The traditional VAE	93.2%
	The federated VAE	91.5%
$F_{3}$	The proposed method	100%
	The traditional VAE	100%
	The federated VAE	100%
$F_{4}$	The proposed method	93.5%
	The traditional VAE	89.2%
	The federated VAE	91.6%

**Table 7 sensors-24-04839-t007:** Performance comparison of different methods in steady operation conditions.

Methods	FAR	Recall	ACR	Average FD Delay
The proposed method	6.8%	96.0%	94.1%	0.00131 s
The traditional VAE	32.4%	94.0%	75.1%	0.00135 s
The federated VAE	8.5%	94.2%	92.3%	0.00132 s

## Data Availability

The datasets presented in this article are not readily available because of the data confidentiality restrictions of CRRC Corporation. Requests to access the datasets should be directed to CRRC Corporation.

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
