# Peer review of "Unsupervised Transfer Learning Method via Cycle-Flow Adversarial Networks for Transient Fault Detection under Various Operation Conditions"

_sensors, 2024, doi:10.3390/s24154839_

Round 1

Reviewer 1 Report

Comments and Suggestions for Authors

The paper employs a migration learning strategy based on federated cyclic flow adversarial networks (CFANs) to diagnose transient faults in the traction system of high-speed trains. The paper is more structured, the method is feasible, and the results of experimental validation are better. However, there are still a few points that require further clarification or elaboration from the authors.

1)The paper does not appear to present an overall flowchart of the methodology introduced therein. It is therefore recommended that the authors add this to facilitate the reader's quick understanding of the overall framework of the paper.

2)The paper introduces a neural network structure similar to adversarial migration. However, it does not provide a detailed discussion on the domain matching problem, the overfitting risk problem, and the migration learning strength problem. It is therefore recommended that a discussion be added on the rationalization reasons for the design of the neural network structure and whether to circumvent the above problems.

3)The authors have only briefly compared the transient fault detection accuracy of several methods in the experimental section. It is recommended that the authors evaluate the neural network performance in several aspects, including accuracy, recall, ROC-AUC metrics, and so forth. Additionally, the model visualization section should be expanded to facilitate a deeper understanding of the decision-making process and internal representation of the model.

Comments on the Quality of English Language

The paper is fluent in English, but part of the language style is not academic enough, and the author is advised to check for further embellishment.

Author Response

Comments 1: The paper does not appear to present an overall flowchart of the methodology introduced therein. It is therefore recommended that the authors add this to facilitate the reader's quick understanding of the overall framework of the paper.

Response 1: The authors thank the reviewer for this suggestion.

The flowchart of the proposed method is illustrated in Figure 8, comprising an offline training phase and an online fault detection (FD) phase. The first CFAN-based model CFAN1 is trained by using the normal data Ds obtained during steady operation conditions to extract latent variables and reconstruct data. Subsequently, the model CFAN2 undergoes federated training based on dynamic operation condition data Dt. The trained federated neural networks CFAN1and CFAN2 enable feature extraction and reconstruction of the healthy data. Thus, the residual r are calculated by the federated CFANs. Finally, With the FD threshold Jth determined by RMS statistics of residual r, the J(r(m))RMS of the testing data are compared with Jth to realize the FD of the TCS. (Figure 8 is shown in the attached file.)

Figure 8. The overall flowchart of the proposed FD method.

Comments 2: The paper introduces a neural network structure similar to adversarial migration. However, it does not provide a detailed discussion on the domain matching problem, the overfitting risk problem, and the migration learning strength problem. It is therefore recommended that a discussion be added on the rationalization reasons for the design of the neural network structure and whether to circumvent the above problems.

Response 2: Thanks for the valuable comments.

The rationalization behind the design of the neural network structure and its ability to address the aforementioned issues, including domain matching problem and migration learning strength problem, are clearly elucidated as follows:

Firstly, source domain data is reconstructed by adversarial learning method, we regard the steady and dynamic state of TCS as the source domain and target domain, respectively. The purpose of the CFAN1 is to learn knowledge from source domain, feature extraction and data reconstruction.Subsequently, the probability density function obtained from the kernel density estimation, as depicted in the figure below, illustrates the distribution characteristics of both dynamic and steady-state operational data for TCS. Despite a certain disparity between the dynamic and steady-state operating conditions, their distributions exhibit similarity. Therefore, the domain matching problem can be solved via transfer learning method.

Afterwards, residual signals caused by the varied domains are incorporated into training CFAN2. As described in Algorithm 2 and formula (31), the CFAN2 adaptively adjusts fault detection performance by residual signal e1(n) to accommodate changes in operation conditions. (Algorithm 2 and formula (31) are shown in the attached file.)

Finally, as demonstrated in Table 4 and Table 5, for weak TFs (cases study F1, F2, F4), this proposed method exhibits superior fault detection capabilities under dynamic operating conditions. The federated CFANs-based transfer learning method applies the knowledge learned from the source domain to the target domain to improve the fault detection performance.

To mitigate the issue of overfitting, gradient vanishing, and gradient exploding, we have simplified proposed model resulting in a lightweight CFAN architecture. As depicted in Table 2, Table 3, and Table 4, despite having only 3200 parameters, CFAN achieves superior FDR values for four faults: 94.2%, 96.2%, 100%, and 93.5% respectively; outperforming both traditional VAEs and federated VAEs. Furthermore, CFAN exhibits better performance across metrics such as FAR, Recall, and ACR when compared to traditional VAEs and federated VAEs (Table 5).

In addition, instead of Stochastic Gradient Descent (SGD), the Adaptive Moment Estimation (ADAM) is adopted, which combines the advantages of momentum and adaptive learning rate. This approach facilitates faster convergence to local optimal solutions while also providing regularization effects to mitigate overfitting. (Table 2, Table 3, Table 4, Table 5 are shown in the attached file.)

Comments 3: The authors have only briefly compared the transient fault detection accuracy of several methods in the experimental section. It is recommended that the authors evaluate the neural network performance in several aspects, including accuracy, recall, ROC-AUC metrics, and so forth. Additionally, the model visualization section should be expanded to facilitate a deeper understanding of the decision-making process and internal representation of the model. 

Response 3: Many thanks for the valuable suggestion. Section 4 has been deeply revised.

In order to facilitate a deeper understanding of the decision-making process and internal representation of the model, in section 3 and 4, the figure 7, 8, 10-16 have been revised thoroughly.

The proposed model has been evaluated by several metrics as bellow:

The training loss curve is defined by Mean Squared Error (MSE) for evaluating reconstruction accuracy. As illustrated in Figure 16(a), during the training of the proposed method, the training loss stabilizes at a lower level than the other two methods, indicating the superior data reconstruction capabilities of the proposed method. Figure 16(b) displays the loss of CFAN2 and federated VAE network2. The losses of two methods converge to similar value, which illustrates that both networks have ability to achieve performance adjustments. (Figure 16(a) is shown in the attached file.)

The ROC-AUC(Receiver Operating Characteristic - Area Under the Curve) curves of three methods are shown in Figure 17, the AUC of the proposed method is 0.953, while the AUC of the traditional VAE and the federated VAE are 0.826 and 0.907, respectively. The proposed method has the largest area under the curve, indicating superior performance in terms of FD. (Figure 17 is shown in the attached file.)

Reviewer 2 Report

Comments and Suggestions for Authors

1.      I hope the authors would organize symbols.

2.      In Equation 3, the authors assumed z probability distribution. I hope the authors would explain that the assumption is reasonable.

3.      The italic expression of mathematical denotements, such as log and det, causes confusion.

4.      There is no explanation for some abbreviations.

5.      I hope the authors would describe the specifications of experimental system.

6.      I hope the authors would describe the limitations (red line) in Figures 9-11.

7.      In Figures 9-11, for the x label, time is better than step size.

8.      I hope the authors would show various case studies.

Comments on the Quality of English Language

the paper is a little hard to read.

I hope the authors would refine a sentences and equations.

Author Response

Comments 1: I hope the authors would organize symbols

Response 1: Thanks for the valuable comments. The consistency in notation has been carefully checked. The symbols are revived accordingly and highlighted in the manuscript. (The instance is shown in the attached file.)

Comments 2: In Equation 3, the authors assumed z probability distribution. I hope the authors would explain that the assumption is reasonable.

Response 2: The author sincerely appreciates the valuable suggestion.

The assumption of z probability distribution is explained clearly in the file submitted.

Comments 3: The italic expression of mathematical denotements, such as log and det, causes confusion.

Response 3: Thanks for the valuable comment.

The italic expression has been thoroughly revised, and highlighted in the manuscript.

Comments 4: There is no explanation for some abbreviations.

Response 4: The abbreviations have been revised and shown in the file submitted.

Comments 5: I hope the authors would describe the specifications of experimental system.

Response 5: Many thanks for the valuable suggestion.

The detailed parameters of the experimental platform have been provided in Table 1. This experimental system is established by Guangdong Railway Vehicles Co., Ltd. (CRRC) (Table 1 is shown in the attached file.)

Comments 6: I hope the authors would describe the limitations (red line) in Figures 9-11

Response 6: The serials of Figures 9-11 have revised into Figures 10-13.

In order to enhance clarity and comprehension, we have provided a detailed description of the thresholds (red lines) depicted in Figures 10-13.

The revised description is as follows:

Comparisons between each FD task and other methods were conducted, encompassing four types of FD tasks and fault-free detection tasks for each method. Figures 10 to 13 illustrate the FD results obtained using both the proposed method and VAE (including transfer and non-transfer learning). The traditional VAE refers to the VAE method without transfer learning, while the federated VAE, which incorporates a similar transfer learning strategy as our proposed method, is adapted for dynamic operating conditions. The red dotted line in the figures represents the FD threshold Jth. (Figures 10-13 are shown in the attached file.)

Comments 7: In Figures 9-11, for the x label, time is better than step size.

Response 7: The authors thank the reviewer for the suggestion. The serials of Figures 9-11 have revised into Figures 10-13. The x-axis of figures have been revised and shown in the attached file.

Comments 8: I hope the authors would show various case studies.

Response 8: We appreciate for your valuable comments sincerely.

The occurrence of the F4 can be attributed to IGBT damage resulting from internal structural defects, manufacturing processes, and other contributing factors. Furthermore, excessive stress induced by high temperatures may lead to gate driver circuit failure, such as TF caused by erroneous pulse control signals originating from the control circuit.

In section 4, the FD task for F4 is conducted, and the relevant metrics and figures have been thoroughly analyzed accordingly as illustrated in Figure 11(d)-Figure 14(d) and Table 4 and Table 5. (Figure 11(d)-Figure 14(d) and Table 4 and Table 5 are shown in the attached file.)

Round 2

Reviewer 1 Report

Comments and Suggestions for Authors

After reading the paper carefully, I think that the author has made significant improvements to the structure and logic of the paper after the revision, the work of the paper proposes valuable and innovative solutions for the field of transient fault monitoring, the methodology proposed in the paper has strong theoretical support, and the analysis of the results is more detailed and clearer, and the readability of the paper as a whole is still good, on the whole, the research work of the paper has a certain value, and I think the paper is worth accepting.

Author Response

Comments 1: After reading the paper carefully, I think that the author has made significant improvements to the structure and logic of the paper after the revision, the work of the paper proposes valuable and innovative solutions for the field of transient fault monitoring, the methodology proposed in the paper has strong theoretical support, and the analysis of the results is more detailed and clearer, and the readability of the paper as a whole is still good, on the whole, the research work of the paper has a certain value, and I think the paper is worth accepting.

Response 1: Thanks indeed for the careful review and positive comments, as well as agree to accept this paper.

Reviewer 2 Report

Comments and Suggestions for Authors

1.      Using the calligraphic for the font style of N in the line 174 is better.

2.      The italic expression of mathematical denotements, such as log and det, causes confusion.

3.      I hope the authors would use some examples to demonstrate the transformation described in Fig. 2.

4.      In the paper, the authors seem to set the threshold using characteristics similar to the standard deviations. In that case, the threshold can be tight, such as causing frequent false alarms. I hope the authors would describe why they set the threshold using this approach and think of other better approaches.

5.      It also connects to comment 4. It seems that the threshold set approach only applies to the intermittent fault type. It seems that the approach cannot be applied to other kinds of faults, such as the abrupt and incipient types.
